# A large planetary body inferred from diamond inclusions in a ureilite meteorite

Farhang Nabiei[1,2], James Badro[1,3], Teresa Dennenwaldt[2,4], Emad Oveisi [2], Marco Cantoni[2], Cécile Hébert[2,4], Ahmed El Goresy[5], Jean-Alix Barrat[6] & Philippe Gillet[1]

Planetary formation models show that terrestrial planets are formed by the accretion of tens of Moon- to Mars-sized planetary embryos through energetic giant impacts. However, relics of these large proto-planets are yet to be found. Ureilites are one of the main families of achondritic meteorites and their parent body is believed to have been catastrophically disrupted by an impact during the first 10 million years of the solar system. Here we studied a section of the Almahata Sitta ureilite using transmission electron microscopy, where large diamonds were formed at high pressure inside the parent body. We discovered chromite, phosphate, and (Fe,Ni)-sulfide inclusions embedded in diamond. The composition and morphology of the inclusions can only be explained if the formation pressure was higher than 20 GPa. Such pressures suggest that the ureilite parent body was a Mercury- to Mars-sized planetary embryo.

[1] Earth and Planetary Science Laboratory (EPSL), Institute of Physics, Ecole Polytechnique Fédérale de Lausanne, Lausanne, Switzerland. [2] Interdisciplinary Center for Electron Microscopy (CIME), Ecole Polytechnique Fédérale de Lausanne, Lausanne, Switzerland. [3] Institut de Physique du Globe de Paris, Sorbonne Paris Cité, Paris, France. [4] Electron Spectrometry and Microscopy Laboratory (LSME), Institute of Physics, Ecole Polytechnique Fédérale de Lausanne, Lausanne, Switzerland. [5] Bayerisches Geoinstitut, Universität Bayreuth, Bayreuth, Germany. [6] Institut Universitaire Européen de la Mer, Université de Bretagne Occidentale, Plouzané, France. Correspondence and requests for materials should be addressed to F.N. (email: farhang.nabiei@epfl.ch)

Asteroid 2008 TC$_3$ fell in 2008 in the Nubian desert in Sudan[1], and the recovered meteorites, called Almahata Sitta, are mostly dominated by ureilites along with various chondrites[2]. Ureilite fragments are coarse grained rocks mainly consisting of olivine and pyroxene, originating from the mantle of the ureilite parent body (UPB)[3] that has been disrupted following an impact in the first 10 Myr of the solar system[3]. High concentrations of carbon distinguishes ureilites from all other achondrite meteorites[3], with graphite and diamond expressed between silicate grains.

There are three mechanisms suggested for diamond formation in ureilites: (i) shock-driven transformation of graphite to diamond during a high-energy impact[4], (ii) growth by chemical vapor deposition (CVD) of a carbon-rich gas in the solar nebula[5], and (iii) growth under static high-pressure inside the UPB[6]. Recent observation[7] of a fragment of the Almahata Sitta ureilite (MS-170) revealed clusters of diamond single crystals that have almost identical crystallographic orientation, and separated by graphite bands. It was thus suggested that individual diamond single crystals as large as 100 μm existed in the sample, which have been later segmented through graphitization[7]. The formation of such large single-crystal diamond grains along with $\delta^{15}$N sector zoning observed in diamond segments[7] is impossible during a dynamic event[8,9] due to its short duration (up to a few seconds[10]), and even more so by CVD mechanisms[11], leaving static high-pressure growth as the only possibility for the origin of the single-crystal diamonds.

Owing to their stability, mechanical strength and melting temperature, diamonds very often encapsulate and trap minerals and melts present in their formation environment, in the form of inclusions. In terrestrial diamonds, this has allowed to estimate the depth of diamond formation, and to identify the composition and petrology of phases sampled at that depth. Therefore, diamonds formed inside the UPB can potentially hold invaluable information about its size and composition.

In this study, we investigated the Almahata Sitta MS-170 section using transmission electron microscopy (TEM) and electron energy-loss spectroscopy (EELS). We studied the diamond–graphite relation and discovered different types of inclusions that were chemically characterized by energy dispersive X-ray (EDX) spectroscopy, crystallographically by electron diffraction, and morphologically by TEM imaging. The composition and mineralogy of these inclusions points to pressures in excess of 20 GPa inside the UPB, which in turn implies a planetary body ranging in size between Mercury and Mars.

## Results

**Diamond–graphite relationship**. The diamond matrix shows plastic deformation as evidenced by the high density of dislocations, stacking faults and a large number of {111} deformation twins (Supplementary Fig. 1). Despite no sign of graphitization for uninterrupted twins, the deformation twins that intersect an inclusion transform to graphite (Fig. 1, Supplementary Fig. 2), while keeping their original morphology. Thus, the diamond–graphite grain boundary forms parallel to the {111} planes of diamond (Supplementary Note 1).

The sample shown in Fig. 2 consists of several diamond segments with close crystallographic orientations, and are separated by graphite bands. Inclusion trails can be seen extending from one diamond segment into the next, while disappearing in the in-between graphite band (Fig. 2b). This is undeniable morphological evidence that the inclusions existed in diamond before these were broken into smaller pieces by graphitization. Similar to the graphitized twins, the graphite bands in Fig. 2 have grain boundaries parallel to {111} planes of diamond

(Supplementary Fig. 3 and Supplementary Note 1). Thus, the most likely cause of graphitization is the shock event where the diamond matrix has been severely deformed[12,13]. Elevated temperature during the shock, as well as stress concentration around the inclusion promotes the graphitization process[13,14].

**Iron–sulfur type inclusions in diamond**. The overwhelming majority of inclusions are iron-rich sulfides, found either as isolated grains with sizes up to a few 100 nanometers, or as trails of small particles ranging from 50 nm down to a few nanometers (Fig. 3 and Supplementary Fig. 4). All the inclusions are faceted indicating that they were trapped as solid crystalline phases rather than melts. However, they show evidence of transformation to low-pressure phases during decompression, similarly to those found in deep terrestrial diamond inclusions[15]. Both chemical and crystallographic analysis (Supplementary Table 1 and Supplementary Fig. 5) show that the sulfide inclusions have dissociated to three phases (Fig. 2c): FeS-troilite, (Fe,Ni)-kamacite, and minor amounts of (Fe,Ni)$_3$P-schreibersite. The latter either dissociates to a separately detectable phosphide phase in larger inclusions (Fig. 3 and Supplementary Fig. 4), or concentrates at grain boundaries in smaller inclusions (Supplementary Fig. 4). It is noteworthy that troilite, kamacite, and schreibersite are never found as isolated mono-mineralic inclusions in the diamonds, but always together inside a very sharply defined polyhedral arrangement; two arguments promoting the idea that these inclusions crystalized as a single-Fe–Ni–S–P phase during diamond formation, that later decomposed into different phases. This is further confirmed by the constant and stoichiometric bulk chemical composition of these inclusions. In order to avoid any sampling bias in such multicomponent inclusions, the composition was measured only on those grains that were completely embedded inside the diamond host determined by electron tomography, leaving aside those that had been partially cut during focused ion beam (FIB) preparation. We found an average molar (Fe + Ni)/(S + P) ratio of 2.98±0.36 from 29 sulfide inclusions (Supplementary Table 2), which corresponds to an (Fe, Ni)$_3$(S,P) initial mineralogy. (Fe,Ni)$_3$P-schreibersite and (Fe,Ni)$_3$S have the same space group (tetragonal I$\bar{4}$) and their lattice parameters are very close[16,17], allowing them to form a solid

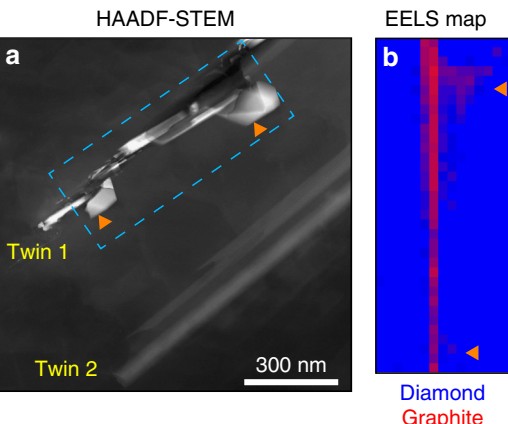

**Fig. 1** Graphitization of diamond along twinning directions. **a** The high-angle annular dark-field (HAADF) STEM image shows two twinning regions indicated as twin 1 and twin 2. Twin 1 is intersecting with two inclusions (indicated by orange arrows) and graphitized, while twin 2 is purely diamond. **b** The graphite-diamond EELS map (from the dashed blue rectangle in panel **a**) indicates that the graphitization is confined to the twinning region and around the inclusions (red = graphite, blue = diamond)

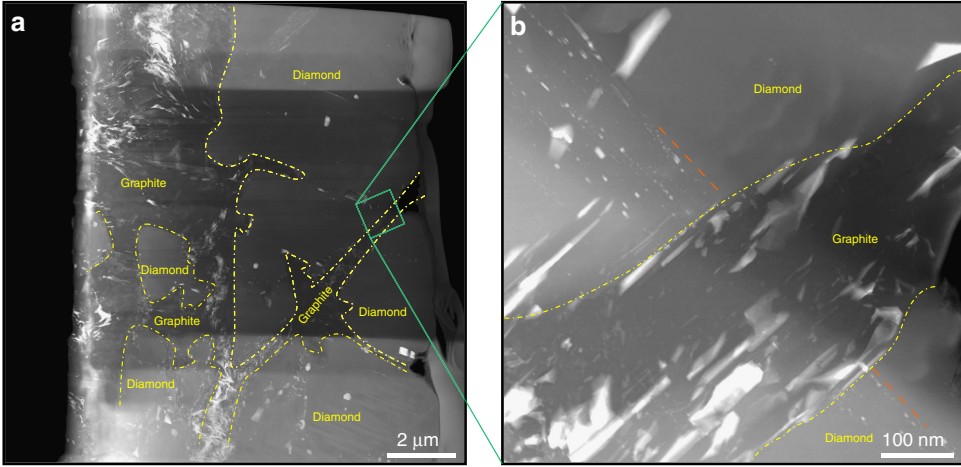

**Fig. 2** Inclusion trails imaged inside diamond fragments. **a** HAADF-STEM image from diamond segments with similar crystallographic orientation. Dashed yellow lines show the diamond–graphite boundaries. **b** High-magnification image corresponding to the green square in **a**. Diamond and inclusion trails are cut by a graphite band. The dashed orange line shows the direction of the inclusion trails

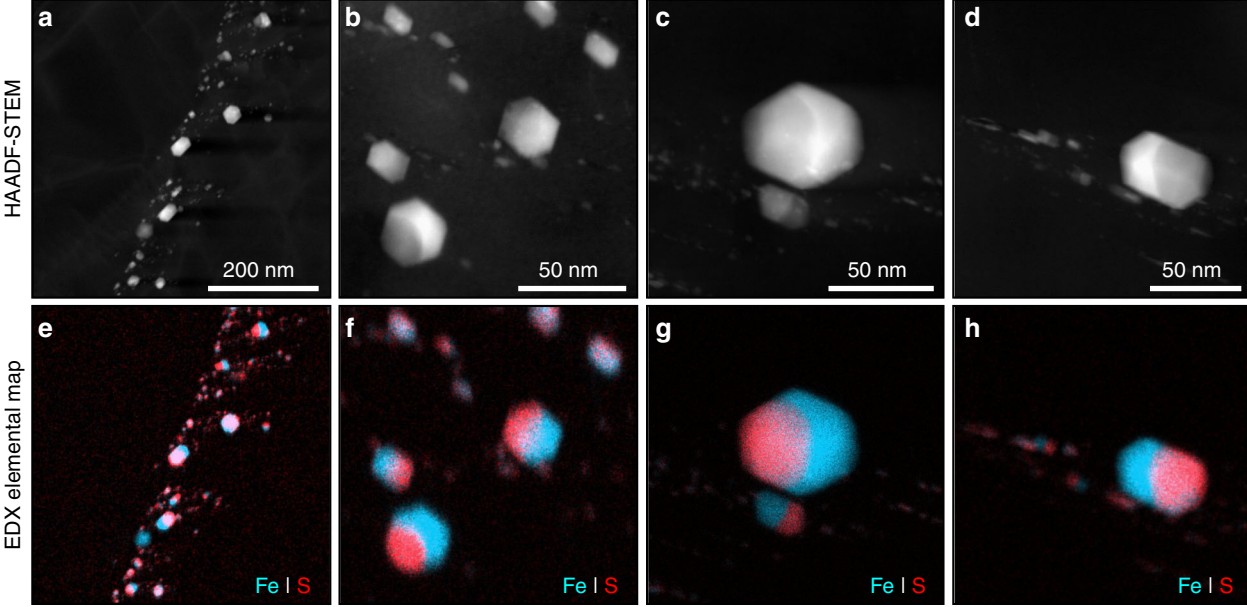

**Fig. 3** Electron micrograph and compositional maps of diamond inclusions in ureilite. HAADF-STEM images (**a**, **b**, **c**, and **d**) and associated Fe and S elemental maps (**e**, **f**, **g**, and **h**) of inclusions in diamond. All chemical (EDX) maps show Fe (light blue) and S (red) distribution. Kamacite and troilite phases appear as light blue and reddish-pink respectively

solution at high pressures as $(Fe,Ni)_3(S,P)$[16,17] across the entire compositional S–P join.

The pressure stability of the $Fe_3(S,P)$ phase depends[18] on its composition (Supplementary Note 2 and Supplementary Fig. 6), and ranges from 21 GPa for the $Fe_3S$ to room pressure for $Fe_3P$, allowing to use the $P/(S + P)$ ratio as an internal thermobarometer. Phosphorus has no effect on the stability for $P/(S + P)$ between 0 and 0.2, $Fe_3(S,P)$ is only stable above 21 GPa[18] (Supplementary Fig. 6) just like $Fe_3S$. The average $P/(S + P)$ of the inclusions observed here is 0.12±0.02 (Supplementary Table 2), and therefore these can only have formed above 21 GPa. Similarly, the inclusions contain nickel, with $Ni/(Fe + Ni)$ = 0.068 ± 0.011, which could also have an effect on the stability pressure of $(Fe,Ni)_3(S,P)$, with $Ni_3S$ (isostructural with $Fe_3S$)[19] stable only above 5.1 GPa. We lack the experimental work to evaluate the pressure effect of Ni substitution for Fe, but assuming a linear dependence of pressure stability on Ni content, the $(Fe, Ni)_3(S,P)$ inclusions would only form above ~20 GPa (Supplementary Note 2 and Supplementary Fig. 7). It is noteworthy that pressure-composition phase diagrams are often concaved downward, and there could be, just as with S–P substitution, no effect on pressure at those low Ni concentrations, so that 20 GPa is actually a lower bound for the inclusions' formation pressure (Supplementary Fig. 7).

**Chromite and phosphate inclusions in diamond**. A second type of inclusions, $Cr_2FeO_4$ chromite, are rare (with only a few identified in the samples) but rather large with grains a few hundred nanometers across (Supplementary Fig. 8). The mineralogy of chromite grains is well preserved and chemical analysis confirms a stoichiometric $Cr_2FeO_4$ chromite (Supplementary Note 3), with no Mg

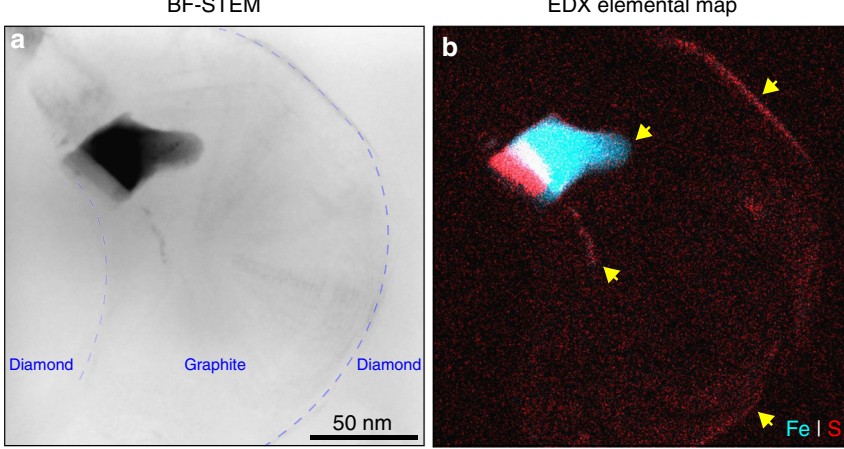

**Fig. 4** Electron micrograph and chemical map of an inclusion in a graphitized region. **a** Bright-field (BF) STEM image and **b** chemical (EDX) map from graphite growth in diamond matrix around an inclusion. Blue dashed lines indicate the diamond–graphite boundary. The yellow arrows point out the Fe–S-rich regions in graphite. Notice the clear rounded form of the inclusion in graphitized part indicating partial melting

or Al substitution for Fe and Cr, respectively. While chromite is often observed in meteorites, Mg- and Al-free end-members are only found in iron meteorites[20–22]. It has been proposed that such end-members must form in a metallic melt with low Cr and O concentration close to the Fe–FeS join[22,23]. Therefore, these chromites must have formed in an iron-rich environment.

Finally, rare Ca–Fe–Na phosphate inclusions were found, roughly ~20 nanometer or smaller (Supplementary Fig. 8), which were only characterized chemically due to their small size (not structurally due to overlap with the surrounding diamond). These inclusions are chemically similar to the ones observed in iron meteorites where they are the most common companions of pure $Cr_2FeO_4$ chromites[24] (Supplementary Note 3).

**Iron–sulfur type inclusions in graphite.** Whereas the polyhedral shapes and consistent bulk composition of inclusions in diamond shows that these phases were a single-homogeneous solid phase at the time of diamond formation, the morphology of inclusions in neighboring graphitized bands shows evidence of melting (Fig. 2a and 4, Supplementary Fig. 9). Indeed, Fe- and S-bearing phases of varying composition and arbitrary shapes are dispersed in the graphitized areas and between graphite layers (Fig. 2a and 4, Supplementary Fig. 9), which provides an evidence for melting of inclusions at the time of graphitization, and yet another indication that graphitization is subsequent to diamond formation. This also provides an explanation for the transformation of original $(Fe,Ni)_3(S,P)$ solid solution to kamacite, troilite and schreibersite phases while keeping the polyhedral shape and bulk composition of the initial parental phase. Graphitization is likely caused by a shock event, which is followed by separation from the parent body and, therefore a pressure drop. That same shock event should melt the inclusions, which then recrystallize after the pressure drop as kamacite, troilite and schreibersite, which are the equilibrium phases at low pressures. The volume change during melting would also add to the strain concentration around them, which in turn facilitates the graphitization process.

## Discussion

The segment sizes of diamonds are not measured in this study; however, the segments we used for sample preparation were all over 10 μm in diameter. Our results also confirm the previous suggestion that the large diamond crystallites are later segmented through graphitization during a shock event. Thus, considering previous studies using electron backscatter diffraction[7], we can conclude that there were diamond grains as large as 100 μm in this particular meteorite. The surprisingly large size of diamond grains and specifically $\delta^{15}N$ sector zoning[7] is incompatible with formation by shock metamorphism. Indeed, laboratory shock experiments are generally done in nanoseconds and natural shocks by impact in the solar system have durations ranging from microseconds up to at most a few seconds[10]. The typical grain size for shock produced diamond is in the order of few nanometers up to few tens of nanometers[8,9,25]. Diamond composite aggregates can reach several hundreds of microns in exceptional cases like Ries and Popigai craters where graphitic precursors are known[9,26]. However, the crystallite size in these aggregates never exceeds 150 nm[8,9,25]. In contrast, the diamond grain size we observe in Almahata Sitta MS-170 samples are 2–4 orders of magnitudes larger than the shock produced diamonds[7]. Such large diamonds are even less likely to grow by CVD in the solar nebula[11]. Moreover, the existence of inclusions in these diamonds and the pressure required to form them (above 20 GPa) clearly rules out the CVD growth mechanism. Therefore, we can distinguish two distinct types of diamond in ureilites: Multigrain diamond resulting from shock events producing clumps of nm-sized individual diamonds[4], and large diamonds up to 100 μm in diameter growing at high-static pressure inside the proto-planet[7] subsequently broken down to equally oriented segments of several tens of micrometer in diameter.

Ureilites are unique samples from the mantle of a differentiated parent body. It has been shown that temperature inside the UPB was higher than the Fe–S eutectic temperature[27,28] (~1250 K at ambient pressure[29], ~1350 K at 21 GPa[16]). Therefore, an Fe-S melt must have percolated and segregated to form a sulfur-bearing metallic core[27,28], but the temperature was never high enough for complete melting of silicates and metallic iron[30], and the core formation process continued until the UPB's mantle reached 20–30 vol% of melt fraction[31].

The composition of chromite inclusions in diamonds shows that they have formed from iron-rich composition without any interaction with silicates. Otherwise, chromite would have accommodated Mg and Al in its composition similarly to the previously reported chromites in ureilite meteorites[32]. This corroborates the formation of the sulfide, chromite, and phosphate inclusions in a metallic liquid.

Moreover, the Fe–C binary system also has a eutectic point (~1400 K at ambient pressure)[33]. Fe–C and Fe–S liquids are immiscible at ambient pressure, but the miscibility gap closes by increasing the pressure above 4–6 GPa (depending on the

composition)[34–36]. Therefore, for a carbon-rich body such as the UPB, we can expect to have a single-Fe–S–C liquid at high pressures. It has been recently shown that large terrestrial diamonds have formed from an Fe–S–C (with Ni and P) liquid[37]. $Fe_3S$ and diamond are the first solids to crystallize (liquidus phases) on the iron-poor side of the Fe–S and Fe–C eutectics, respectively; it is therefore likely that they can simultaneously crystallize from a cooling Fe–S–C liquid above 20 GPa inside the UPB. Although an experimental study of the Fe–S–C ternary system is required to examine this possibility, the distribution of iron–sulfur inclusions in the diamonds supports this idea. The arrangement of small inclusions in vein-like trails (Fig. 2) is consistent with the formation from a liquid phase at the same time or immediate aftermath (depending on the UPB's thermal history) of the solidification of the UPB, rather than from the transformation of graphite to diamond at depth. This is corroborated by the widespread distribution of $(Fe,Ni)_3(S,P)$ inclusions in diamond which is unlikely to take place by diffusion inside a graphitic precursor.

There is considerable debate on the size of the UPB[3,38,39]. A body of at least ~1000 km in diameter was recently suggested to account for the pressure required to form diamond (above 2 GPa) in the depths of its mantle[7]. Here we show that these diamonds contain inclusions that can only form above ~20 GPa, which can only be attained in a large planetary body. If the diamonds formed at the core-mantle boundary, the UPB would be Mars-sized. The lower-bound for its size is for them to form at the center of the UPB, and a 20 GPa center is consistent with a Mercury-sized body.

Although this is the first compelling evidence for such a large body that has since disappeared, their existence in the early solar system has been predicted by planetary formation models[40]. Moon- to Mars- sized planetary embryos have formed either by runaway[41] and oligarchic growth[42] of planetesimals or by pebble accretion[43] in the first million years of the solar system. Mars-sized bodies (such as the giant impactor that formed the Moon[44,45]) were common[43], and either accreted to form larger planets, or collided with the Sun or were ejected from the solar system[46,47]. This study provides convincing evidence that the ureilite parent body was one such large "lost" planet before it was destroyed by collisons[48].

## Methods

**Focused ion beam sample preparation.** Samples for TEM investigations were prepared using the conventional in situ lift-out technique in a Zeiss NVision 40 dual beam instrument. The polished surface of the MS-170 section from Almahata Sitta meteorite was coated with ~15 nm carbon to increase the conductivity during FIB milling. After identifying the target diamond grains (secondary electron detector at 5 kV), they were coated with ~2 μm amorphous carbon (ion beam induced deposition) in order to protect the interesting area during the ion milling. The diamond grain was milled with $Ga^+$ ions at 30 kV starting with 27 nA current and going down to 700 pA, until we obtained a ~1 μm thick slice. This slice was then transferred and attached to a cupper grid with a carbon deposition. To make the slice electron transparent (to ~100 nm in thickness), it was thinned down with low-beam currents (ranging from 700 pA down to 80 pA). At the end the slice was polished with $Ga^+$ ions at 5 kV and 2 kV using 30 pA and 25 pA beam currents, respectively. Five thin sections were prepared for transmission electron microscopy (TEM) studies.

**Energy electron loss spectroscopy.** EELS analysis was performed on a FEI Titan Themis TEM operated at 80 kV. The carbon K-edge was recorded by electron spectroscopic imaging (ESI) in scanning TEM (STEM) mode with dual-channel EELS for near-simultaneous low-loss and core-loss acquisition with a dispersion of 0.1 eV/channel, an entrance aperture of 2.5 mm and a camera length of 115 mm resulting in a convergence angle of 3.78 mrad and a collection semi-angle of 5.1 mrad satisfying the magic angle condition (MAC). The MAC allows the determination of the ratio of $sp^2/sp^3$ bonding in carbon (R-ratio) independent of specimen orientation[49,50]. The spectrum fitting method which uses either Gaussian or Lorentzian (or the combination of both) functions to fit the peaks accounting for the π\* and σ\* states was applied to determine the R-ratio maps. The $sp^2/sp^3$ ratio of a highly oriented pyrolytic graphite (HOPG) was used as a reference to normalize the meteorite R-ratio maps. A channel-to-channel gain variation and dark current correction were done for all EEL spectra. This allows concluding that the carbon

phases in presence are either pure cubic diamond or pure graphite. Then, reference spectra are obtained for pure diamond ($S_D$) and pure graphite ($S_G$). Each pixel spectrum ($S_{Px}$) from the EELS maps is linearly fitted with the two reference spectra as below:

$$S_{px} = k_1 S_G + k_2 S_D.$$

The graphite/(diamond + graphite) ratio is obtained as $k_1/(k_2 + k_1)$. This has been illustrated as RGB map (Fig. 1) where red, green, and blue are corresponding to the graphite ratio, zero, and the diamond ratio, respectively.

**Weak-beam imaging and electron diffraction.** The weak-beam dark-field imaging technique was used to observe dislocations and stacking faults in the diamond. This technique allows the observation of defects with sharper contrast compared to the background. We first tilted the specimen to satisfy a systematic row two-beam diffraction condition (direct beam and **g** reflection excited). From this setting, the beam was tilted slightly to excite the 3**g** reflection. The **g** reflection was selected by the objective aperture to acquire a weak-beam dark-field (**g**,3**g**) image, whose signal is very sensitive to deformation fields around the dislocation core and stacking faults. This imaging technique, as well as electron diffraction analysis were conducted on a FEI Tecnai Osiris machine at 200 kV. Nano-diffraction was done using the smallest condenser aperture and largest spot-size in "nano-beam" mode. Although, the resulting beam was not completely parallel, the diffraction spots were sharp and small enough for accurate indexing.

**STEM imaging and EDX analysis.** STEM imaging and EDX analysis were performed on a FEI Tecnai Osiris microscope at 200 kV. This microscope is equipped with four window-less silicon drift (SSD) EDX detectors and Esprit 1.9 acquisition software from Bruker.

The large effective area of the 4 detectors significantly increases the count rate of photons. However, it also suffers from shadowing effects that might affect the accuracy of the quantification. To avoid that, quantification was done on the EDX maps acquired at 20° sample tilt and only the two detectors facing the sample (the other two detectors were switched off). To determine the Fe/S ratio a troilite reference in equilibrium with kamacite inside an inclusion was used. The identity of troilite was confirmed by electron diffraction. The measured error was below 4%.

**STEM tomography.** Tilt series of high-angle annular dark-field (HAADF) STEM images were acquired form different regions of the section. The HAADF detector had a collection angle larger than 63.8 mrad or 100.1 mrad, corresponding to camera lengths of 91 mm and 58 mm, respectively, in order to reduce the contribution of diffraction contrast in the images. The electron beam convergence angle was set to 10 mrad in order to increase the depth of focus.

Tilt series were acquired using a tomography sample holder (Fischione model 2040) on a FEI Titan Themis microscope operated at 300 kV. Large magnification series were obtained from −72 to 72 degrees with a step size of 2 degrees. This was used to observe the faceted shapes of inclusions. Then, several other tilt series acquisitions were obtained at lower magnification from −54 to 54 degrees with 2 degree intervals. The purpose of these tilt series was to identify the position of inclusions inside the diamond matrix. These identified uncut inclusions were taken for EDX quantification. All reconstructions and visualizations were done using the Inspect3D and Chimera software packages, respectively.

**Data availability.** The data that support the findings of this study are available from the corresponding author upon reasonable request.

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

## Acknowledgements

We thank Dr. Duncan T. L. Alexander for his assistance regarding the EELS acquisition. We also thank Fabienne Bobard for her help during the sample preparation with FIB. We are grateful to Dr. Richard Gaal for the scientific discussions about this research. This work was supported by the Swiss National Science Foundation through FND grant 200021_140474.

## Author contributions

F.N. and P.G. planned the research. F.N. performed STEM imaging and EDX analysis. F. N. and M.C. prepared the sample with FIB. F.N., T.D., and C.H. obtained the EELS data and analyzed them. F.N. and E.O. did the weak-beam imaging and electron tomography. F.N., M.C., and C.H. analyzed the microscopy data. J.B., A.E., J.-A.B., and P.G. provided scientific support and contributed to the discussions. F.N. and J.B. wrote the paper. P.G. supervised the whole project. All the authors contributed to the revisions.

## Additional information

**Competing interests:** The authors declare no competing interests.

