## [Peer Review File · Nature Communications]

Reviewers' comments:

Reviewer #1 (Remarks to the Author):

This is a very interesting paper, with potentially far-reaching consequences. Unfortunately, some of the central arguments are either incorrect or at least not clearly explained.

Line 85: "Mg- and Al-free end-members are only found in the metal phases of pallasites and iron meteorites." Not true. Such chromites are found in metamorphosed carbonaceous chondrites, for example. Therefore the succeeding assertion, that "It has been proposed that such end-members must form from a metallic melt with low Cr and O concentration close to the Fe-FeS join. Therefore, these chromites must have formed in the vicinity of the UPB's core." is incorrect, and cannot be used as evidence for the paper's central contention.

Figure 2: If P is present in all inclusions mentioned, why is it only visible in one panel of the figure?

Line 116: The authors of reference 26 state that they expect that Fe₃(S,P) could be formed by shock, so a static high pressure appears to not be required, despite the argument presented in lines 120-130.

Therefore I cannot recommend publication of the current paper. If a revised manuscript is requested by the Editor, I recommend that a native English speaker provide a final edit, as there are numerous minor grammar errors in the current manuscript.

Reviewer #2 (Remarks to the Author):

Nabiei et al. present evidence that the large (<100 um) diamonds found in the ureillite (a type of primitive, partially differentiated yet carbon-rich meteorite) named Almhata Sitta enclose Fe,Ni-S,P-rich inclusions of homogenous chemical composition (as well as chromite without Mg, Al additions, indicating their likely formation in a Fe,S-rich melt). Nabiei et al. suggest that these minerals are the low-pressure polymorphs of another phase originally formed at a pressure of about 21 GPa. In combination with previously reported N-zoning found in these diamonds, which supports in-situ formation and thus a static pressure >2 GPa, this observation favors the formation of these diamonds in a large planetary body (21 GPa corresponds to the pressure at the core-mantle boundary of Mars).

This claim is certainly novel: while large radii for the ureillite parent body (UPB) have been suggested before (Miyahara et al. 2015 suggested >2 GPa and >1000 km diameter; Goodrich et al. 2004, *Chemie der Erde* 64:283 determined, but did not advocate, that a potential disruption of the UPB at the time of the late heavy bombardment period at 3.8 Ga would mean that the UPB was about lunar-sized), this is the first study to provide evidence of formation pressures only expected in planets / planetary embryos.

The size of the UPB has previously been estimated to be around 250 km by other authors (e.g., Wilson et al. 2008, *Geochimica et Cosmochimica Acta* 72:6154), but this size estimate is model-dependent (it requires equilibrium smelting to establish the mg-content of olivines in different ureillites, which is controversial, e.g., Warren et al., 2006 *Geochimica et Cosmochimica Acta* 70:2104). If the claims provided by Nabiei et al. are confirmed, this would therefore also have significant consequences for currently existing models of petrogenesis of the ureillites (e.g., the "full-smelting" model would have to be discarded). It would also likely trigger a search for similar inclusions in other types of meteorites, and lead to a better estimation of the stability fields of the Fe-Ni-S-P system. If ureillites are indeed fragments from a large, "lost" planet, this would provide us with a unique insight into the planet formation processes in the early solar system.

The article is easy to read and understand and well documented (as far as I can tell - I am not a crystallographer or experimental mineralogist). I would however recommend the authors to be somewhat more cautious with that one number (21 GPa) from which their inferred UPB size crucially depends. First of all, as the authors admit themselves (which I applaud), the consequences of adding Ni to the phase diagram are not yet clear (they have to "assume ideality"). Then, it should be noted that the phase diagram provided by Gu et al. 2016, *American Mineralogist* 101:205 is given for a temperature of 900-1000°C. However, several different mineral thermometers yield consistently higher values of 1200-1300°C for the temperatures at which the ureilites were quenched (presumably during the catastrophic disruption of the UPB), see e.g. Singletary & Grove 2003, *Meteoritics & Planetary Science* 38:95. These higher temperatures would also require some reconciliation with the suggested crystallinity of the inclusions given a eutectic of ~1100°C (at 21 GPa). I think it would be good if at least some first-order estimate of the uncertainty of that value (21 GPa) and its "translation" into UPB sizes could be provided by the authors. This might then also be used to support the "Mercury- to Mars-sized" description for the size of the UPB given in the abstract (but currently not explained further in the text).

These ureilite mineral temperatures provide another potential challenge to the idea of a "Mars-sized" UPB: temperatures at the CMB of such a large planet are thought to be much higher, perhaps around 1700°C (see e.g. Ruedas et al. 2013 *Physics of the Earth and Planetary Interiors* 216:32). In addition, the UPB has been shown to have formed while primordial ²⁶Al was still active (based on excess ²⁶Mg; Kita et al. 2003, *Lunar & Planetary Science Conference* 34, 1557), i.e., within a few Ma after formation of the first condensates in the solar system - such an early formation would certainly lead to additional heating of the body from radioactive decay. Also, Warren et al. 2006 find that ureilite show signs of a removal of a Fe,S-rich melt, but almost no depletion in Fe or siderophile elements - how would this be reconciled with presumably rapid core-formation in such a large planet given a relatively strong gravity/buoyancy? Therefore, I find the scenario of a Mars-sized UPB currently difficult to reconcile with existing observations. While it would be excessive to ask the authors to fully resolve these potential problems, I would at least expect them to mention the ureilite mineral thermometers, and perhaps also shortly discuss how the temperature-pressure-conditions documented by these diamond inclusions might potentially be reconciled with the conditions expected in planetary interiors (e.g., an even larger planet might reach 21 GPa closer to the surface, where temperatures are lower? A heterogenous distribution of ²⁶Al? A substantial ice-fraction for the UPB, providing excess pressure but not much heating? Did the diamonds originate in another object and just mixed into the UPB through a collision?).

I am also a bit surprised that the authors cite Miyahara et al. 2015 as a source for the claim that formation by vapor deposition in the solar nebula can be excluded for the large Almahata Sitta diamonds. Much to the contrary, Miyahara et al. suggest that vapor deposition is one of two possible formation scenarios (even though such large diamonds have yet to be observed in interstellar space or circumstellar environments). Instead, the argumentation should be exactly the other way around: the high pressures recorded by the inclusions in these diamonds (as shown bei Nabiei et al. in the present paper) clearly refute the idea that they formed under low-pressure conditions in the solar nebula! I also think that the formation of "diamonds" in ureilites likely requires more than one formation mechanism (as various authors have suggested), which should be clarified somewhere in the text.

Matthias M. M. Meier

Reviewer #3 (Remarks to the Author):

The manuscript NCOMMS-17-15212-T reported diamond inclusions in Ureilites meteorite, which might give new insight into the formation and evolution of the parent body that has not been preserved during the early history of solar system. The discovery of Fe-Ni-S-P bearing inclusions

with consistent (Fe+Ni)/(S+P) ratio and phosphate inclusions in Ureilites meteorite is relatively novel, which could be interesting for readers in Earth science field. However, the current data and evidence might not be solid enough to support the conclusions, and the author needs to clarify certain statements in the paper to make it more intelligible.

Fe₃S and Fe₃(S,P) phases can form under high pressure within eutectic temperature, and they are quenchable phases as shown by previous experimental studies (Ref 8, 26). However, it looks that the author didn't observed any grains with both S and P alloyed with Fe as Fe₃(S,P) solid solution and it would be important for the author to discuss the reason. Failing to form this solid solution would mainly due to either a not high enough pressure or a high temperature condition that leads to the decomposition of the phase as shown by the phase diagram (Ref 8). As the authors favor the later situation, they should provide more evidence and discussions to strengthen the idea that the diamond has undergone high temperature event that causes the decomposition (which might happen during the shock event when the meteorite entering the Earth), and the discussion of the formation condition of other coexisting inclusions and diamond deformation features would also be helpful to constrain the thermal condition experienced by the meteorite.

Line 52-61 The author needs to better explain why the inclusions exist in diamond before they were cut by graphitization. The morphological feature showing here is relatively complex, and it would be clearer if the author can explain the implication of each morphological feature one by one (dislocation, deformation twins, morphological of the inclusion), so that it would be easier to understand the conclusion in this paragraph.

Line 52 As for the dislocation and deformation twins, do they indicate certain temperature/pressure conditions? Please add discussion about this issue.

Fig 1b. Is that the same area as Fig 1a? It's unclear about the diamond and graphite boundary on Fig 1b. It's better to mark it on the figure or add the info in the caption.

Line 87 Info of ref 22 is not complete. Does the coexisting of chromite and phosphate phases indicate certain thermal conditions? It would be better to discuss the forming conditions of chromite and phosphate phases to better constrain the growth conditions of Fe-Ni-S-P phase. In line 135, the author mentioned the depletion of Mg and Al, but it's unclear which reference was mentioned here for the statement. In Ref 21, chromite was found within the system Fe-Cr-S-O in Saint Aubin meteorite, whose type is different from Ureilite. It's unclear how the author made the conclusion that "these chromites must have formed in the vicinity of the UPB's core" from the evidence of a different meteorite.

Line 88 why "the existence of tiny phosphate inclusions close to the chromites" strengthen the statement that "chromites must have formed in the vicinity of the UPB's core"? The author needs to provide a more comprehensive explanation for the statement.

Figure 2 It would be better if the author can provide separate element maps to show the element distribution as supplementary material (such as Fig 2h). It's unclear how does Fe distribute in the S-rich area from current figures. Extended Data Figure 4 shows separate element maps, but Fe and S maps are not separate.

Line 105 The author gives the molar ratio of (Fe+Ni)/(S+P). It could be more convincing if the author can provide the list of compositional results for Fe, Ni, S, P and (Fe+Ni)/(S+P) ratio of each grains (30 in total) as a table in the supplementary material.

Line 119 Did the author check any area such as Fig 2h to see if there is any Fe₃(S,P) phase? It looks that there is a crossover with S and P map. Is there any point measurement of the composition on that area?

Line 120 I think the author would like to say the $(\text{Fe}+\text{Ni})/(\text{S}+\text{P})$ ratio is steady on each polyhedral shapes of inclusions. It's better to change "homogeneous" to "consistent" to make it less confusing.

Line 133 It's unclear what pressure the author mentioned here for the core-mantle boundary of protoplanets.

Figure 3 line 147 The author mentioned "partial melting" on the blur area of the figure. Is there any high-resolution image on that area showing melting textures?

Response to the reviewers

Reviewer comments are marked in blue.

Our response to each comment or question is given below it in black color.

The correction applied to address the question or the comment is given in green color after our response.

Reviewer #1 (Remarks to the Author):

This is a very interesting paper, with potentially far-reaching consequences. Unfortunately, some of the central arguments are either incorrect or at least not clearly explained.

Line 85: 1C;Mg- and Al-free end-members are only found in the metal phases of pallasites and iron meteorites. 1D; Not true. Such chromites are found in metamorphosed carbonaceous chondrites, for example. Therefore the succeeding assertion, that 1C;It has been proposed that such end-members must form from a metallic melt with low Cr and O concentration close to the Fe-FeS join. Therefore, these chromites must have formed in the vicinity of the UPB 19;s core. 1D; is incorrect, and cannot be used as evidence for the paper 19;s central contention.

This is a very important comment and we want to make sure we are addressing it properly and clearly. We thank the reviewer for pointing out this misunderstanding, as our text may have been misleading. On one hand, we could not find published reports of Mg- and Al-free chromite in carbonaceous chondrites. To the best of our knowledge, Mg- and Al-free chromites are only found in irons and pallasites. On the other hand, the reviewer is perfectly correct: the presence of such chromite minerals in our samples cannot be taken as evidence for the formation of diamonds in the UPB's core. The mineralogical message from these inclusions is that they formed in a Mg- and Al-free environment, which by extension, means a silicate-free or rock-free environment, leaving only one possibility which is that of a metallic environment of formation. Indeed, chromites formed in the presence of Mg and Al will always accommodate these elements in their composition, as has been shown over the years and reported many times in the literature. And conversely, chromites found in iron meteorites or in the metallic phase of pallasites are systematically Mg- and Al-free. As an example, previously reported chromite grains in ureilites have Fe/(Fe+Mg) ranging from ~0 up to ~0.6 and Cr/(Cr+Al) ranging from ~0.5 up to ~0.8¹. Therefore, the chromites seen here are undoubtedly formed with no interaction with silicates, and likely from a Fe-S melt. However, we do agree that this melt is not necessarily in the core; it could very well be melt pockets of core-forming liquid on their way to the core, but not necessarily in the core. Note that if pressures of 20 GPa are reached before even reaching the core, then our argument for a sizeable planet holds all the better. We realize our formulation may

have been misleading so we reformulated the text. This has been corrected in lines 120-121 and added explanations in lines 171-175.

L. 120-121 now reads: Therefore, these chromites must have formed in an iron-rich environment.

L. 171-175 The composition of chromite inclusions in diamonds shows that they have formed from iron-rich composition without any interaction with silicates. Otherwise, chromite would have accommodated Mg and Al in its composition similarly to the previously reported chromites in ureilite meteorites¹. This corroborates the formation of the sulfide, chromite and phosphate inclusions is a metallic liquid

Figure 2: If P is present in all inclusions mentioned, why is it only visible in one panel of the figure?

Phosphorous is present in all the inclusions. However, since its concentration is low compared to Fe and S, including phosphorous EDX maps makes renders the images fuzzy (due to the background noise). This was particularly important for the smaller inclusions that had smaller Schreibersite (Fe,Ni)₃P region. So we clarified all this by doing the following:

Now all images in the main text show only Fe and S EDX maps (Fig. 3). We added Supplementary Figure 4 that shows HAADF image and separated Fe, S, Ni, and P maps for inclusions in 6 different regions, so that it is clear that P is everywhere. Moreover, the explanation of P distribution inside the inclusion can now be found in the manuscript lines 81-84 as well as in the supplementary information line 57-61:

L. 85-88 now reads: ... and minor amounts of (Fe,Ni)₃P-schreibersite. The latter either dissociates to a separately detectable phosphide phase in larger inclusions (Fig. 3 and Supplementary Fig. 4), or concentrates at grain boundaries in smaller inclusions (Supplementary Fig. 4).

supplementary text, L. 57-61: Moreover, we observed a minor phase with the composition 52.2 atomic% Fe, 22.3% Ni, and 25.4% P which corresponds to schreibersite; the phase was easily distinguishable and individualized in larger grains (Supplementary Fig. 4), but smaller inclusions only exhibit P and Ni enrichments at the interfaces between the diamond matrix and the inclusion, or between troilite and kamacite within the inclusions (Supplementary Fig. 4).

We have also added Supplementary Table 2 that shows the detailed composition for 29 intact inclusions.

Line 116: The authors of reference 26 state that they expect that Fe₃(S,P) could be formed by

shock, so a static high pressure appears to not be required, despite the argument presented in lines 120-130.

Reference 18 (previously ref. #26) (Gu et al. 2016²) experiments describe the phase diagram and ***shows*** that Fe₃(S, P) can be formed above 21 GPa. It also ***suggests*** that Fe₃(S, P) could be formed by shock in meteorites, despite the fact that Fe₃(S, P) phase has never been found in any natural samples including highly shock meteorites. (Important to note the lexicological difference between the two highlighted verbs above). So, on the basis of the phase diagram of Fe₃(S, P), indeed, static pressure is not required and one can't discriminate between a shock and static high pressure. All we can say based on Gu et al. is that the composition of the inclusions in our studies indicate a formation pressure above 21 GPa.

Now, the strong evidence against the shock formation of inclusions comes from the diamond growth mechanism. As indicated by Miyahara et al. 2016 and discussed with more evidences in our manuscript and supplementary information, the large diamonds observed in Almahata Sitta MS-170 meteorite cannot be formed during the shock (or by CVD growth). Therefore, since the diamonds are formed in static high-pressure conditions inside the parent body, the inclusions trapped in these diamonds should have been crystallized before or at the same time as diamond formation, which in turn negates their formation by shock.

But we realize again this can be misleading, and that this important part of the discussion was in Supp. Info. and not directly available to the reader. The complete discussion on the diamond formation mechanism was transferred from supplementary materials to the main text line 144-164. Moreover, we have edited the manuscript to clarify the logic for the formation of inclusion inside the parent body based on the diamond formation condition in lines 47-49 of the manuscript. This was also addressed in supplementary information lines 114-120.

L. 144-164 was moved from Supp info into main text and reads: The segment sizes of diamonds are not measured in this study; however, the segments we used for sample preparation were all over 10 μm in diameter. Our results also confirm the previous suggestion that the large diamond crystallites are later segmented through graphitization during a shock event. Thus, considering previous studies using electron backscatter diffraction³, we can conclude that there were diamond grains as large as 100 μm in this particular meteorite. The surprisingly large size of diamond grains and specifically δ¹⁵N sector zoning³ is incompatible with formation by shock metamorphism. Indeed, laboratory shock experiments are generally done in nanoseconds and natural shocks by impact in the solar system have durations ranging from microseconds up to at most a few seconds⁴. The typical grain size for shock produced diamond is in the order of few nanometers up to few tens of nanometers⁵⁻⁷. Diamond composite aggregates can reach several hundreds of microns in exceptional cases like Ries and Popigai craters where graphitic precursors

are known^{6,8}. However, the crystallite size in these aggregates never exceeds 150 nm⁵⁻⁷. In contrast, the diamond grain size we observe in Almahata Sitta MS-170 samples are 2 to 4 orders of magnitudes larger than the shock produced diamonds³. Such large diamonds are even less likely to grow by CVD in the solar nebula⁹. Moreover, the existence of inclusions in these diamonds and the pressure required to form them (above 20 GPa) clearly rules out the CVD growth mechanism. Therefore, we can distinguish two distinct types of diamond in ureilites: Multigrain diamond resulting from shock events producing clumps of nm-sized individual diamonds¹⁰, and large diamonds up to 100 μm in diameter growing at high static pressure inside the proto-planet³ subsequently broken down to equally oriented segments of several tens of micrometer in diameter.

L. 47-49 now reads: Therefore, diamonds formed inside the UPB can potentially hold invaluable information about its size and composition.

Supp Info. L. 114-120 reads: These sulfide inclusions could be syngenetic (i.e. forming at the same time as diamond), or protogenetic (i.e. forming before their encapsulation in diamond)¹¹. Since these inclusions form trails in the diamond matrix, they could not have been subjected to a significant convection before diamond formation and their encapsulation. This supports the hypothesis that the diamond formed at the same time as the inclusions, or at least that the diamond were formed in the aftermath of inclusion formation, at the same place and condition that the inclusions had formed, namely in static high pressure condition of at least 20 GPa inside a planetary body.

Therefore, I cannot recommend publication of the current paper. If a revised manuscript is requested by the Editor, I recommend that a native English speaker provide a final edit, as there are numerous minor grammar errors in the current manuscript.

We wish to thank the reviewer for his/her important comments. We hope we mitigated the concerns by providing a clearer explanation, and by circumventing any misleading statements from the text.

Reviewer #2 (Remarks to the Author):

Nabiei et al. present evidence that the large (<100 μm) diamonds found in the ureilite (a type of primitive, partially differentiated yet carbon-rich meteorite) named Almahata Sitta enclose Fe,Ni-S,P-rich inclusions of homogenous chemical composition (as well as chromite without Mg, Al additions, indicating their likely formation in a Fe,S-rich melt). Nabiei et al. suggest that these minerals are the low-pressure polymorphs of another phase originally formed at a pressure of about 21 GPa. In combination with previously reported N-zoning found in these diamonds, which supports in-situ formation and thus a static pressure >2 GPa, this observation favors the formation of these diamonds in a large planetary body (21 GPa corresponds to the pressure at the

core-mantle boundary of Mars).

This claim is certainly novel: while large radii for the ureilite parent body (UPB) have been suggested before (Miyahara et al. 2015 suggested >2 GPa and >1000 km diameter; Goodrich et al. 2004, *Chemie der Erde* 64:283 determined, but did not advocate, that a potential disruption of the UPB at the time of the late heavy bombardment period at 3.8 Ga would mean that the UPB was about lunar-sized), this is the first study to provide evidence of formation pressures only expected in planets / planetary embryos.

The size of the UPB has previously been estimated to be around 250 km by other authors (e.g., Wilson et al. 2008, *Geochimica et Cosmochimica Acta* 72:6154), but this size estimate is model-dependent (it requires equilibrium smelting to establish the mg-content of olivines in different ureilites, which is controversial, e.g., Warren et al., 2006 *Geochimica et Cosmochimica Acta* 70:2104). If the claims provided by Nabiei et al. are confirmed, this would therefore also have significant consequences for currently existing models of petrogenesis of the ureilites (e.g., the "full-smelting" model would have to be discarded). It would also likely trigger a search for similar inclusions in other types of meteorites, and lead to a better estimation of the stability fields of the Fe-Ni-S-P system. If ureilites are indeed fragments from a large, "lost" planet, this would provide us with a unique insight into the planet formation processes in the early solar system.

The article is easy to read and understand and well documented (as far as I can tell - I am not a crystallographer or experimental mineralogist). I would however recommend the authors to be somewhat more cautious with that one number (21 GPa) from which their inferred UPB size crucially depends. First of all, as the authors admit themselves (which I applaud), the consequences of adding Ni to the phase diagram are not yet clear (they have to "assume ideality"). Then, it should be noted that the phase diagram provided by Gu et al. 2016, *American Mineralogist* 101:205 is given for a temperature of 900-1000°C. However, several different mineral thermometers yield consistently higher values of 1200-1300°C for the temperatures at which the ureilites were quenched (presumably during the catastrophic disruption of the UPB), see e.g. Singletary & Grove 2003, *Meteoritics & Planetary Science* 38:95. These higher temperatures would also require some reconciliation with the suggested crystallinity of the inclusions given a eutectic of $\sim 1100^\circ\text{C}$ (at 21 GPa). I think it would be good if at least some first-order estimate of the uncertainty of that value (21 GPa) and its "translation" into UPB sizes could be provided by the authors. This might then also be used to support the "Mercury- to Mars-sized" description for the size of the UPB given in the abstract (but currently not explained further in the text).

- With respect to Ni

This is absolutely correct: Ni content can change the formation pressure of the $(\text{Fe, Ni})_3(\text{S, P})$ inclusions. Since the Ni content was small, we didn't worry about it, but if a reader (such as Dr. Meier) thinks it can be an issue, then it needs to be addressed!

Unfortunately, there is no high-pressure experimental data on the quaternary system Fe-Ni-S-P. So what we can do is look at the ternaries and try to build our way up to the quaternary. We know that Fe_3S and Ni_3S are stable above at 21 GPa¹² and 5.1 GPa¹³, respectively. A linear approximation puts a formation pressure of 19.9 GPa (~ 20 GPa) for the composition of ureilites studied here with $\text{Ni}/(\text{Fe}+\text{Ni}) = 0.07$. Now, considering that the phase lines should be concave (negative second derivative), this is a lower-bound, and the stability is therefore at least 20 GPa, but potentially more. The absolute value of derivative close to Fe-rich end member is expected to be lower than the absolute value of the slope in linear approximation. Therefore, 1 GPa error is the upper limit of the variation in the formation pressure.

L104-112 now reads: Similarly, the inclusions contain nickel, with $\text{Ni}/(\text{Fe}+\text{Ni})=0.068 \pm 0.011$ which could also have an effect on the stability pressure of $(\text{Fe,Ni})_3(\text{S,P})$, with Ni_3S (isostructural with Fe_3S ¹³) stable only above 5.1 GPa. We lack the experimental work to evaluate the pressure effect of Ni substitution for Fe, but assuming a linear dependence of pressure-stability on Ni content, the $(\text{Fe,Ni})_3(\text{S,P})$ inclusions would only form above ~ 20 GPa (Supplementary Note 2 and Supplementary Fig. 7). It is noteworthy that pressure-composition phase diagrams are often concaved downward, and there could be, just as with S–P substitution, no effect on pressure at those low Ni concentrations, so that 20 GPa is actually a lower bound for the inclusions' formation pressure (Supplementary Fig. 7).

Supp. Info. L100-108 now reads: The Ni content can also affect the formation pressure of $(\text{Fe,Ni})_3(\text{S,P})$ phase. Ni_3S forms above 5.1 GPa¹³. However, experimental data for the formation pressure of $(\text{Fe, Ni})_3\text{S}$ solid solution is not available. Therefore, we have used two end-members, Fe_3S and Ni_3S , to linearly approximate the variation of formation pressure for the $(\text{Fe}_{1-x}, \text{Ni}_x)_3\text{S}$ phase (Supplementary Fig. 7). For $\text{Ni}/(\text{Fe}+\text{Ni}) = 0.07$ this yields the formation pressure of 19.9 GPa (Supplementary Fig. 7). Considering that the real variation of formation pressure for such systems are often concaved downward (negative second derivative) as schematically shown in Supplementary Fig. 7, the value obtained from the linear approximation indicates the lower limit of the formation pressure for this composition.

Supp Figure 7: shows the extrapolation of the phase diagram using Fe_3S and Ni_3S endmembers.

- With respect to temperature

The experiments from Gu et al. 2016² are in lower temperature than expected for the UPB as mentioned by reviewer. However, the Fe-S phase diagram from Fei et al. 2000¹² is the result of experiments at higher temperatures. They show no pressure-dependence on temperature of formation of Fe_3S . Again, all we can do here is suppose that low Ni and P concentrations don't

change that behavior, and there is nothing else we can do. It seems to us that 1 GPa error is acceptable.

Mars- sized body is deduced assuming the formation of inclusions in the mantle. However, if the inclusions are forming in the core, we expect to have smaller parent body. 20 GPa is close to the pressure at the center of a Mercury-sized planet (although not Mercury itself because it has a huge core and a tiny mantle, here we are referring to something more like Earth or Mars with 1/3 core and 2/3 silicate, anyway, this is only a ball-park figure), and a little smaller than pressure at the core-mantle boundary of Mars. Thus, we mentioned a Mercury- to Mars- sized body, and there is no way we can be more accurate than that.

We have added the error estimation based on Ni content in the main text line 104-112 (see above). Also, the plot for linear approximation and its explanations are added to supplementary information in supplementary figure 7 and lines 100-108. Moreover, we have used pressure 20 GPa (instead of 21 GPa) throughout the text to account for the error. But this does not change the general conclusions.

These ureilite mineral temperatures provide another potential challenge to the idea of a "Mars-sized" UPB: temperatures at the CMB of such a large planet are thought to be much higher, perhaps around 1700°C (see e.g. Ruedas et al. 2013 *Physics of the Earth and Planetary Interiors* 216:32). In addition, the UPB has been shown to have formed while primordial ²⁶Al was still active (based on excess ²⁶Mg; Kita et al. 2003, *Lunar & Planetary Science Conference* 34, 1557), i.e., within a few Ma after formation of the first condensates in the solar system - such an early formation would certainly lead to additional heating of the body from radioactive decay. Also, Warren et al. 2006 find that ureilite show signs of a removal of a Fe,S-rich melt, but almost no depletion in Fe or siderophile elements - how would this be reconciled with presumably rapid core-formation in such a large planet given a relatively strong gravity/buoyancy? Therefore, I find the scenario of a Mars-sized UPB currently difficult to reconcile with existing observations. While it would be excessive to ask the authors to fully resolve these potential problems, I would at least expect them to mention the ureilite mineral thermometers, and perhaps also shortly discuss how the temperature-pressure-conditions documented by these diamond inclusions might potentially be reconciled with the conditions expected in planetary interiors (e.g., an even larger planet might reach 21 GPa closer to the surface, where temperatures are lower? A heterogenous distribution of ²⁶Al? A substantial ice-fraction for the UPB, providing excess pressure but not much heating? Did the diamonds originate in another object and just mixed into the UPB through a collision?).

We thank the reviewer for this comment, and indeed, we see this study as a first step to identify the nature and size of disrupted planetary embryos in the early solar system. We are thankful for the suggestions and have used them in our conclusion to open up discussion and foster additional interest in the study of ureilites.

The temperature of equilibration of silicates in ureilites is estimated to be around 1200-1300 °C. And as mentioned by the reviewer the ureilite has been partially differentiated, but it did not go through the complete differentiation. The exact temperature of the UPB and its differentiation condition not only depends on the formation time and the size but also on the formation place in the solar system and the rate of its growth¹⁴. This is especially challenging for ureilite meteorite which show properties that cannot be easily matched with each other. For instance, while lithophile elemental composition¹⁵ and stable isotopic ($\epsilon^{62}\text{Ni}$, $\epsilon^{50}\text{Ti}$, and $\epsilon^{54}\text{Cr}$) composition¹⁶ undeniably point towards the UPB in inner Solar System, the oxygen isotopic composition and high carbon and volatile concentrations are compatible with the formation beyond the snow line¹⁵ (which increases the ice fraction as you have mentioned and can also lead to the fast formation of very large planetary bodies). Also, the carbon-oxygen isotopic dichotomy points to two distinct reservoirs in the UPB, once again inconsistent with full-scale planetary differentiation. Moreover, the formation mechanism itself can change the picture about the temperature in the large planetary bodies. For example pebble accretion (models that are developed for giant planets, but they are also getting attention to explain the formation of terrestrial planets) can potentially deposit less energy on the growing body compared to the classical runaway and oligarchic growth. Also, as you mentioned, the accretion of materials from two different parent bodies is a possibility in ureilites. Thus, simply we cannot reconcile and explain the complete thermal history of these meteorites. However, the composition of inclusions found in these ureilite diamonds gives new insight about the differentiation process in the UPB. We have added a part in the paper to suggest the formation of diamonds and inclusions from the segregated S-bearing iron melt. We also mentioned that this could come from the impactor, and then that would be the large planetary embryo, not the UPB. But then, what are ureilites and what is the UPB? This could be a mixture of two bodies. All this is clearly beyond the scope of this paper, and we hope we addressed partially these concerns. This also means that they can potentially open a door to further explore the samples from earliest stages of the planetary differentiation. We hope that this study gives new insights and opens new discussions in this subject.

We have added three paragraphs in discussion section of the manuscript (L. 165-189) to discuss the possible implication of our data for the differentiation process in the UPB. We have also added a sentence in L. 202-204 concerning the possibility of two distinct origins for ureilite materials.

L. 165-189 Ureilites are unique samples from the mantle of a differentiated parent body. It has been shown that temperature inside the UPB was higher than the Fe-S eutectic temperature^{17,18} (~1250 K at ambient pressure¹⁹, ~1350 K at 21 GPa¹²). Therefore, an Fe-S melt must have percolated and segregated to form a sulfur-bearing metallic core^{17,18}, but the temperature was never high enough for complete melting of silicates and metallic iron²⁰, and the core formation process continued until the UPB's mantle reached 20 to 30 vol% of melt fraction²¹.

The composition of chromite inclusions in diamonds shows that they have formed from iron-rich composition without any interaction with silicates. Otherwise, chromite would have

accommodated Mg and Al in its composition similarly to the previously reported chromites in ureilite meteorites¹. This corroborates the formation of the sulfide, chromite and phosphate inclusions in a metallic liquid.

Moreover, the Fe-C binary system also has a eutectic point (~1400 K at ambient pressure)²². Fe-C and Fe-S liquids are immiscible at ambient pressure, but the miscibility gap closes by increasing the pressure above 4-6 GPa (depending on the composition)²³⁻²⁵. Therefore, for a carbon-rich body such as the UPB, we can expect to have a single Fe-S-C liquid at high pressures. It has been recently shown that large terrestrial diamonds have formed from an Fe-S-C (with Ni and P) liquid²⁶. Fe₃S and diamond are the first solids to crystallize (liquidus phases) on the iron-poor side of the Fe-S and Fe-C eutectics, respectively; it is therefore likely that they can simultaneously crystallize from a cooling Fe-S-C liquid above 20 GPa inside the UPB. Although an experimental study of the Fe-S-C ternary system is required to examine this possibility, the distribution of iron-sulfur inclusions in the diamonds supports this idea. The arrangement of small inclusions in vein-like trails (Fig. 2) is consistent with the formation from a liquid phase, rather than from the transformation of graphite to diamond at depth. This is corroborated by the widespread distribution of (Fe,Ni)₃(S,P) inclusions in diamond which is unlikely to take place by diffusion inside a graphitic precursor.

L. 202-204 Ureilites would then be the fragments of this body, or a mixture of the fragments of the UPB and its impactor, as suggested by the ureilite carbon-oxygen isotopic dichotomy²⁷.

I am also a bit surprised that the authors cite Miyahara et al. 2015 as a source for the claim that formation by vapor deposition in the solar nebula can be excluded for the large Almahata Sitta diamonds. Much to the contrary, Miyahara et al. suggest that vapor deposition is one of two possible formation scenarios (even though such large diamonds have yet to be observed in interstellar space or circumstellar environments). Instead, the argumentation should be exactly the other way around: the high pressures recorded by the inclusions in these diamonds (as shown by Nabiei et al. in the present paper) clearly refute the idea that they formed under low-pressure conditions in the solar nebula! I also think that the formation of "diamonds" in ureilites likely requires more than one formation mechanism (as various authors have suggested), which should be clarified somewhere in the text.

Although Miyahara et al. 2015³ suggested that the CVD growth of these diamonds is unlikely, but as Dr. Meier suggested they did not refute it. In fact, we have cited "Raty, J.-Y. & Galli, G. Ultradispersity of diamond at the nanoscale. *Nat. Mater.* **2**, 792–795 (2003)" which argues that the size of CVD grown diamond from nebula cannot be larger than nanometric-scale. But it is exactly true that the inclusions in diamond and the pressure required for their formation provides much stronger logic to refute the CVD growth from nebula.

The shock-induced origin of diamonds is the most widely accepted model for diamond formation in ureilites and it is valid for many studied ureilite samples¹⁰. However, the reasons provided for

this mechanism in previous studies are simply not valid for the diamonds studied here. Therefore, as Dr. Meier pointed out there are two type of diamonds found in ureilite meteorite. This has been clarified in the text.

The reason for refutation of CVD growth mechanism is added in line 157-160 of the manuscript. Also, the existence of two diamond type formed through different mechanisms is discussed in lines 160-164 of the manuscript.

L. 157-164 now read: Such large diamonds are even less likely to grow by CVD in the solar nebula⁹. Moreover, the existence of inclusions in these diamonds and the pressure required to form them (above 20 GPa) clearly rules out the CVD growth mechanism. Therefore, we can distinguish two distinct types of diamond in ureilites: Multigrain diamond resulting from shock events producing clumps of nm-sized individual diamonds¹⁰, and large diamonds up to 100 μm in diameter growing at high static pressure inside the proto-planet³ subsequently broken down to equally oriented segments of several tens of micrometer in diameter.

Reviewer #3 (Remarks to the Author):

The manuscript NCOMMS-17-15212-T reported diamond inclusions in Ureilites meteorite, which might give new insight into the formation and evolution of the parent body that has not been preserved during the early history of solar system. The discovery of Fe-Ni-S-P bearing inclusions with consistent (Fe+Ni)/(S+P) ratio and phosphate inclusions in Ureilites meteorite is relatively novel, which could be interesting for readers in Earth science field. However, the current data and evidence might not be solid enough to support the conclusions, and the author needs to clarify certain statements in the paper to make it more intelligible.

Fe₃S and Fe₃(S,P) phases can form under high pressure within eutectic temperature, and they are quenchable phases as shown by previous experimental studies (Ref 8, 26). However, it looks that the author didn't observed any grains with both S and P alloyed with Fe as Fe₃(S,P) solid solution and it would be important for the author to discuss the reason. Failing to form this solid solution would mainly due to either a not high enough pressure or a high temperature condition that leads to the decomposition of the phase as shown by the phase diagram (Ref 8). As the authors favor the later situation, they should provide more evidence and discussions to strengthen the idea that the diamond has undergone high temperature event that causes the decomposition (which might happen during the shock event when the meteorite entering the Earth), and the discussion of the formation condition of other coexisting inclusions and diamond deformation features would also be helpful to constrain the thermal condition experienced by the meteorite.

The melting of Fe-S type inclusion can be deduced from the morphology of these inclusions in the graphitized regions. As indicated in figure 4 and supplementary figure 9, the Fe-S materials are dispersed inside the graphitized region in arbitrary shapes, whereas they have regular polyhedral shapes in diamond. This is a first indication that they melted during graphitization. Moreover, in graphitized regions, kamacite and troilite are separated into two distinct regions. And this is not observed anywhere in the diamond. Molten inclusions inside the diamond keep their original shape, while those in the graphite are spread during the shock induced graphitization process that requires instability in the diamond lattice and displacement of the crystallographic planes.

Since pressure release occurs before temperature release in a shock event, solidification of Fe-S inclusions takes place at low pressure (likely <1 GPa). Thus, the liquid crystallizes as the equilibrium phases at low pressure conditions which are kamacite, troilite, and schreibersite. At this pressure conditions schreibersite ((Fe,Ni)₃P) is not expected to accommodate sulfur in its structure.

The manuscript has been edited to address the melting of inclusions in L. 135-140. Also, the discussion in L. 128-142 as well as figure 4 and supplementary figure 9 clarified the morphology of iron-sulfur materials inside the graphite which cannot arise from solid dispersion.

L. 127-141 now read: Whereas the polyhedral shapes and consistent bulk composition of inclusions in diamond shows that these phases were a single homogeneous solid phase at the time of diamond formation, the morphology of inclusions in neighboring graphitized bands shows evidence of melting (Fig. 2a and 4, Supplementary Fig. 9). Indeed, Fe- and S-bearing phases of varying composition and arbitrary shapes are dispersed in the graphitized areas and between graphite layers (Fig. 2a and 4, Supplementary Fig. 9), which provides an evidence for melting of inclusions at the time of graphitization, and yet another indication that graphitization is subsequent to diamond formation. This also provides an explanation for the transformation of original (Fe,Ni)₃(S,P) solid solution to kamacite, troilite and schreibersite phases while keeping the polyhedral shape and bulk composition of the initial parental phase. Graphitization is likely caused by a shock event, which is followed by separation from the parent body and, therefore a pressure drop. That same shock event should melt the inclusions, which then recrystallize after the pressure drop as kamacite, troilite and schreibersite which are the equilibrium phases at low pressures. The volume change during melting would also add to the strain concentration around them, which in turn facilitates the graphitization process.

Line 52-61 The author needs to better explain why the inclusions exist in diamond before they were cut by graphitization. The morphological feature showing here is relatively complex, and it would be clearer if the author can explain the implication of each morphological feature one by one (dislocation, deformation twins, morphological of the inclusion), so that it would be easier to understand the conclusion in this paragraph.

To better clarity we have separated the previous figure 1 into two figures and explained each figure separately. In three out of five samples studied here we have seen that the Fe-S inclusions are arranged in localized vein-like trails. In sample 4 (figure 2), we can see these trails that are elongated in one diamond segment and they disappear inside the graphite region. Then, they appear again in the next diamond segment in the same direction. Therefore, we have concluded that these trails existed inside a large diamond matrix (similar to other samples) and later they have been cut by graphitized band. This graphitized band divides the diamond and the inclusion trails into two segments. This is also supported by the similar crystallographic orientation of the diamond segments in two sides of the graphite band.

The previous figure 1 has been divided to two figures and each one is explained separately. In figure 2 we have the low and high magnification images from the same sample which can clarify the morphological features there. These explanations are in the manuscript line 57-72.

L. 59-75: The diamond matrix shows plastic deformation as evidenced by the high density of dislocations, stacking faults and a large number of $\{111\}$ deformation twins (Supplementary Fig. 1). Despite no sign of graphitization for uninterrupted twins, the deformation twins that intersect an inclusion transform to graphite (Fig. 1, Supplementary Fig. 2), while keeping their original morphology. Thus, the diamond-graphite grain boundary forms parallel to the $\{111\}$ planes of diamond (Supplementary Note 1).

The sample shown in Fig. 2 consists of several diamond segments with close crystallographic orientations, and are separated by graphite bands. Inclusion trails can be seen extending from one diamond segment into the next, while disappearing in the in-between graphite band (Fig. 2b). This is undeniable morphological evidence that the inclusions existed in diamond before these were broken into smaller pieces by graphitization. Similar to the graphitized twins, the graphite bands in Fig. 2 have grain boundaries parallel to $\{111\}$ planes of diamond (Supplementary Fig. 3 and Supplementary Note 1). Thus, the most likely cause of graphitization is the shock event where the diamond matrix has been severely deformed^{28,29}. Elevated temperature during the shock as well as stress concentration around the inclusion promotes the graphitization process^{29,30}.

Line 52 As for the dislocation and deformation twins, do they indicate certain temperature/pressure conditions? Please add discussion about this issue.

The presence of mechanical twins and dislocation in the diamond matrix show large plastic deformation above the diamond yield strength. These deformation features point to high strain rates that are consistent with the one expected from a shock event (added to Supplementary Information L. 8-9). However, no additional constraint can be implied from the deformation at this point.

Supplementary Information L. 8-9 Moreover, these deformation features point to high strain rates that are consistent with the one expected from a shock event.

Fig 1b. Is that the same area as Fig 1a? It is unclear about the diamond and graphite boundary on Fig 1b. It is better to mark it on the figure or add the info in the caption.

Previous figure 1a and 1b are acquired from two different samples. These images both point out to the graphitization after the formation of diamond. They depict the graphitization of diamond.

We have separated these figures to new figure 1 and figure 2.

Line 87 Info of ref 22 is not complete. Does the coexisting of chromite and phosphate phases indicate certain thermal conditions? It would be better to discuss the forming conditions of chromite and phosphate phases to better constrain the growth conditions of Fe-Ni-S-P phase. In line 135, the author mentioned the depletion of Mg and Al, but it is unclear which reference was mentioned here for the statement. In Ref 21, chromite was found within the system Fe-Cr-S-O in Saint Aubin meteorite, whose type is different from Ureilite. It is unclear how the author made the conclusion that 1C; these chromites must have formed in the vicinity of the UPB core 1D; from the evidence of a different meteorite.

The complete citation is: Ulf-Møller, F. Solubility of Chromium and Oxygen in Metallic Liquids and the Co-Crystallization of Chromite and Metal in Iron Meteorite Parent Bodies. *29th Lunar and Planetary Science conference*, abstract no. 1969, Houston (1998). There is no specific temperature constraint. It is suggested that “chromite crystallizes in the Fe-Cr-S-O system from liquids with very low amounts of Cr and O close to the Fe-FeS join”.

Indeed, we cannot make the conclusion that these chromites are formed in the vicinity of the UPB core. This was already addressed by reviewer 1. However, the chromite will accommodate Mg and Al if it was formed through any interaction with silicates. Composition of the chromites previously reported in ureilite have Fe/(Fe+Mg) ranging from ~0 up to ~0.6 and Cr/(Cr+Al) ranging from ~0.5 up to ~0.8¹. The Al- Mg- free end-member that is reported here can only form in iron-rich environment. The chromite in iron meteorites are forming from Fe-S liquid with small Cr and O content. Similarly, we have suggested that the chromite inclusions observed in our samples are forming from S-bearing iron melt that has been shown to segregate in the UPB^{17,18}.

The manuscript has been corrected to suggest the formation of chromite and inclusions in an iron-rich environment instead of core vicinity. Please see comment for Reviewer 1, reprinted below:

L. 120-121 now reads: Therefore, these chromites must have formed in an iron-rich environment.

L. 171-175 The composition of chromite inclusions in diamonds shows that they have formed from iron-rich composition without any interaction with silicates. Otherwise, chromite would have accommodated Mg and Al in its composition similarly to the previously reported chromites in ureilite meteorites¹. This corroborates the formation of the sulfide, chromite and phosphate inclusions is a metallic liquid

Line 88 why 1C;the existence of tiny phosphate inclusions close to the chromites 1D; strengthen the statement that 1C;chromites must have formed in the vicinity of the UPB 19;s core 1D;? The author needs to provide a more comprehensive explanation for the statement.

Unfortunately, we cannot determine the crystallography and exact composition of these phosphates (please see supplementary information line 138-143 and supplementary figure 8), this is at the limits of what we can resolve and we are using one of the most highly resolved TEMs in the world. Therefore, we cannot deduce constraints about their formation condition, nor apply the same mechanics as for the other inclusions. All we can say with respect to phosphates is qualitative, and our sole aim is to show that the little that can be said is in accord with the rest of what can be obtained from Fe-S and chromites, and not in contradiction. The qualitative composition (Ca-Fe phosphate sometimes with Na) is similar to the phosphates found in iron meteorites. That alone is insufficient to claim that their observation strengthens the formation in core vicinity, as reviewer #3 pointed out. So we have removed the sentence.

The manuscript is corrected in lines 122-126 to address the issue mentioned by reviewer #3. Also, the discussion about the formation condition of these inclusions is added in lines 177-181 of the manuscript.

L122-126 Lastly, rare Ca-Fe-Na phosphate inclusions were found, roughly ~20 nanometer or smaller (Supplementary Fig. 8), which were only characterized chemically due to their small size (not structurally due to overlap with the surrounding diamond). These inclusions are chemically similar to the ones observed in iron meteorites where they are the most common companions of pure Cr₂FeO₄ chromites³¹ (Supplementary Note 3).

L. 171-175: The composition of chromite inclusions in diamonds shows that they have formed from iron-rich composition without any interaction with silicates. Otherwise, chromite would have accommodated Mg and Al in its composition similarly to the previously reported chromites in ureilite meteorites¹. This corroborates the formation of the sulfide, chromite and phosphate inclusions is a metallic liquid.

Figure 2 It would be better if the author can provide separate element maps to show the element

distribution as supplementary information (such as Fig 2h). It is unclear how does Fe distribute in the S-rich area from current figures. Extended Data Figure 4 shows separate element maps, but Fe and S maps are not separate.

The separate EDX maps are added as Supplementary figure 4.

Line 105 The author gives the molar ratio of (Fe+Ni)/(S+P). It could be more convincing if the author can provide the list of compositional results for Fe, Ni, S, P and (Fe+Ni)/(S+P) ratio of each grains (30 in total) as a table in the supplementary information.

This is added as Supplementary Table 2.

Line 119 Did the author check any area such as Fig 2h to see if there is any Fe₃(S,P) phase? It looks that there is a crossover with S and P map. Is there any point measurement of the composition on that area?

The TEM images and EDX maps show the 2-dimensional projection of 3-dimensional objects. Therefore, the overlap between S and P maps are arising due to this 2-D projection. Through the EDX measurements (including point measurements) and electron diffraction analysis we did not find any evidence of additional phases.

Line 120 I think the author would like to say the (Fe+Ni)/(S+P) ratio is steady on each polyhedral shapes of inclusions. It is better to change 1C;homogeneous 1D; to 1C;consistent 1D; to make it less confusing.

Thank you very much. The word “consistence” sounds much better.

Edited in line 128 of the manuscript.

Line 133 It's unclear what pressure the author mentioned here for the core-mantle boundary of protoplanets. 28;

Considering the formation of the inclusions at deepest setting of the mantle, the core mantle boundary should have the pressure of at least 20 GPa.

L. 192-194 now reads: Here, we show that these diamonds contain inclusions that can only form above ~20 GPa, which can only be attained in a large planetary body. Assuming the diamonds formed in its deepest setting at the core-mantle boundary, its size is consistent with that of Mars.

Figure 3 line 147 The author mentioned 1C;partial melting 1D; on the blur area of the figure. Is there any high-resolution image on that area showing melting textures?

Since the dispersion happens inside the un-oriented graphite, the high resolution imaging will not reveal much more evidence. However, this region is a graphitized area of up to 200 nm in

diameter inside the diamond matrix. Therefore, we can easily compare the texture and distribution of chemical element in the graphitized area with the inclusions in diamond matrix. The arbitrary shapes of these inclusions in graphite (also in supplementary figure 9) and the change in the shape of inclusions (faceted in diamond and round in graphite) indicates their melting.

References:

1. Goodrich, C. A. *et al.* Petrology of chromite in ureilites: Deconvolution of primary oxidation states and secondary reduction processes. *Geochim. Cosmochim. Acta* **135**, 126–169 (2014).
2. Gu, T., Fei, Y., Wu, X. & Qin, S. Phase stabilities and spin transitions of Fe₃(S_{1-x}P_x) at high pressure and its implications in meteorites. *Am. Mineral.* **101**, 205–210 (2016).
3. Miyahara, M. *et al.* Unique large diamonds in a ureilite from Almahata Sitta 2008 TC3 asteroid. *Geochim. Cosmochim. Acta* **163**, 14–26 (2015).
4. Gillet, P. & El Goresy, A. Shock Events in the Solar System: The Message from Minerals in Terrestrial Planets and Asteroids. *Annu. Rev. Earth Planet. Sci.* **41**, 257–285 (2013).
5. Le Guillou, C., Rouzaud, J. N., Remusat, L., Jambon, A. & Bourot-Denise, M. Structures, origin and evolution of various carbon phases in the ureilite Northwest Africa 4742 compared with laboratory-shocked graphite. *Geochim. Cosmochim. Acta* **74**, 4167–4185 (2010).
6. Ohfuji, H. *et al.* Natural occurrence of pure nano-polycrystalline diamond from impact crater. *Sci. Rep.* **5**, 14702 (2015).
7. Langenhorst, F., Shafranovsky, G. I., Masaitis, V. L. & Koivisto, M. Discovery of impact diamonds in a Fennoscandian crater and evidence for their genesis by solid-state transformation. *Geology* **27**, 747–750 (1999).
8. Goresy, A. E. *et al.* In situ discovery of shock-induced graphite-diamond phase transition in gneisses from the Ries Crater, Germany. *Am. Mineral.* **86**, 611–621 (2001).

9. Raty, J.-Y. & Galli, G. Ultradispersity of diamond at the nanoscale. *Nat. Mater.* **2**, 792–795 (2003).
10. Lipschutz, M. E. Origin of Diamonds in the Ureilites. *Science* **143**, 1431–1434 (1964).
11. Taylor, L. A., Anand, M. & Promprated, P. Diamonds and their inclusions: are the criteria for syngeneses valid. in *8th International Kimberlite Conference. Long Abstract, Victoria, Canada* (Citeseer, 2003).
12. Fei, Y., Li, J., Bertka, C. M. & Prewitt, C. T. Structure type and bulk modulus of Fe₃S, a new iron-sulfur compound. *Am. Mineral.* **85**, 1830–1833 (2000).
13. Urakawa, S., Matsubara, R., Katsura, T., Watanabe, T. & Kikegawa, T. Stability and bulk modulus of Ni₃S, a new nickel sulfur compound, and the melting relations of the system Ni-NiS up to 10 GPa. *Am. Mineral.* **96**, 558–565 (2011).
14. Šrámek, O., Milelli, L., Ricard, Y. & Labrosse, S. Thermal evolution and differentiation of planetesimals and planetary embryos. *Icarus* **217**, 339–354 (2012).
15. Goodrich, C. A. *et al.* Origin and history of ureilitic material in the solar system: The view from asteroid 2008 TC3 and the Almahata Sitta meteorite. *Meteorit. Planet. Sci.* **50**, 782–809 (2015).
16. Warren, P. H. Stable isotopes and the noncarbonaceous derivation of ureilites, in common with nearly all differentiated planetary materials. *Geochim. Cosmochim. Acta* **75**, 6912–6926 (2011).
17. Warren, P. H., Ulff-Møller, F., Huber, H. & Kallemeyn, G. W. Siderophile geochemistry of ureilites: A record of early stages of planetesimal core formation. *Geochim. Cosmochim. Acta* **70**, 2104–2126 (2006).
18. Barrat, J. A. *et al.* Early stages of core segregation recorded by Fe isotopes in an asteroidal mantle. *Earth Planet. Sci. Lett.* **419**, 93–100 (2015).

19. Ryzhenko, B. & Kennedy, G. C. The effect of pressure on the eutectic in the system Fe-FeS. *Am. J. Sci.* **273**, 803–810 (1973).
20. Greenwood, R. C., Franchi, I. A., Jambon, A. & Buchanan, P. C. Widespread magma oceans on asteroidal bodies in the early Solar System. *Nature* **435**, 916–918 (2005).
21. Guan, Y. & Crozaz, G. Microdistributions and petrogenetic implications of rare earth elements in polymict ureilites. *Meteorit. Planet. Sci.* **36**, 1039–1056 (2001).
22. Lord, O. T., Walter, M. J., Dasgupta, R., Walker, D. & Clark, S. M. Melting in the Fe–C system to 70 GPa. *Earth Planet. Sci. Lett.* **284**, 157–167 (2009).
23. Corgne, A., Wood, B. J. & Fei, Y. C- and S-rich molten alloy immiscibility and core formation of planetesimals. *Geochim. Cosmochim. Acta* **72**, 2409–2416 (2008).
24. Dasgupta, R., Buono, A., Whelan, G. & Walker, D. High-pressure melting relations in Fe–C–S systems: Implications for formation, evolution, and structure of metallic cores in planetary bodies. *Geochim. Cosmochim. Acta* **73**, 6678–6691 (2009).
25. Deng, L., Fei, Y., Liu, X., Gong, Z. & Shahar, A. Effect of carbon, sulfur and silicon on iron melting at high pressure: Implications for composition and evolution of the planetary terrestrial cores. *Geochim. Cosmochim. Acta* **114**, 220–233 (2013).
26. Smith, E. M. *et al.* Large gem diamonds from metallic liquid in Earth’s deep mantle. *Science* **354**, 1403–1405 (2016).
27. Barrat, J.-A., Sansjofre, P., Yamaguchi, A., Greenwood, R. C. & Gillet, P. Carbon isotopic variation in ureilites: Evidence for an early, volatile-rich Inner Solar System. *Earth Planet. Sci. Lett.* **478**, 143–149 (2017).
28. Chacham, H. & Kleinman, L. Instabilities in Diamond under High Shear Stress. *Phys. Rev. Lett.* **85**, 4904–4907 (2000).

29. He, H., Sekine, T. & Kobayashi, T. Direct transformation of cubic diamond to hexagonal diamond. *Appl. Phys. Lett.* **81**, 610–612 (2002).
30. Harris, J. W. & Vance, E. R. Induced graphitisation around crystalline inclusions in diamond. *Contrib. Mineral. Petrol.* **35**, 227–234 (1972).
31. Olsen, E. J. *et al.* The phosphates of IIIAB iron meteorites. *Meteorit. Planet. Sci.* **34**, 285–300 (1999).

Reviewers' comments:

Reviewer #1 (Remarks to the Author):

The revised manuscript, and the reply to my review provided by the authors, have assuaged my main concerns, and I can now support publication of the paper in Nature.

Reviewer #2 (Remarks to the Author):

I am happy with the changes the authors have made to the paper, and think that most of my comments have been addressed. There are two points, however, that I think need to be looked at again:

A more minor point is that although the authors explain in their answer / rebuttal how the "Mercury-to-Mars-size" mentioned in the abstract is derived (Mercury = if inclusions formed in the core; Mars = inclusions formed at the CMB), they only explain the Mars-size/CMB-connection in the text. For the reader to fully grasp where the "Mercury-to-Mars-size" comes from, it would be important to explain the Mercury-size/center-of-the-core-connection as well.

The somewhat more major point is that, unless I misunderstood something, the temperature at which ureilites were quenched (1200-1300°) is ABOVE the temperature at which the inclusion material (Fe,Ni)₃(S,P) melts at ~20 GPa. On the other hand, the observation of crystal faces suggests that they were solid, not liquid. So are we to expect that the temperature further increased after diamond formation / entrapment of the inclusion? Or that the disrupting collision led to an additional temperature increase? I am certain that there are a few ways that this apparent (?) inconsistency could be addressed, but addressed it should be.

After that, I have no further objections to the publication of this paper.

Response to the reviewers

Reviewer comments are marked in blue.

Our response to each comment or question is given below it in black color.

The correction applied to address the question or the comment is given in green color after our response.

Reviewer #1 (Remarks to the Author):

The revised manuscript, and the reply to my review provided by the authors, have assuaged my main concerns, and I can now support publication of the paper in Nature.

Thanks a lot for your support.

Reviewer #2 (Remarks to the Author):

I am happy with the changes the authors have made to the paper, and think that most of my comments have been addressed. There are two points, however, that I think need to be looked at again:

A more minor point is that although the authors explain in their answer / rebuttal how the "Mercury-to-Mars-size" mentioned in the abstract is derived (Mercury = if inclusions formed in the core; Mars = inclusions formed at the CMB), they only explain the Mars-size/CMB-connection in the text. For the reader to fully grasp where the "Mercury-to-Mars-size" comes from, it would be important to explain the Mercury-size/center-of-the-core-connection as well.

Thank you for mentioning this. It is now addressed in the manuscript line 196-199.

L. 196- 199: If the diamonds formed at the core-mantle boundary, the UPB would be Mars-sized. The lower-bound for its size is for them to form at the center of the UPB, and a 20 GPa center is consistent with a Mercury-sized body.

The somewhat more major point is that, unless I misunderstood something, the temperature at which ureilites were quenched (1200-1300°) is ABOVE the temperature at which the inclusion material (Fe,Ni)₃(S,P) melts at ~20 GPa. On the other hand, the observation of crystal faces suggests that they were solid, not liquid. So are we to expect that the temperature further increased after diamond formation / entrapment of the inclusion? Or that the disrupting collision

led to an additional temperature increase? I am certain that there are a few ways that this apparent (?) inconsistency could be addressed, but addressed it should be.

The 1200-1300 °C is the high temperature record in UPB which has been deduced from equilibration of silicate grains¹. The maximal temperature in the UPB could have been even higher than this as evidenced by melting of silicates material². However, this temperature condition is uncorrelated to the temperature constraint set by the formation of diamonds and their inclusions.

The UPB was cooled down after the solidification of the mantle. In fact, ureilites show evidence of rapid cooling from temperatures of about 1050-1100 °C. This temperature range is slightly lower than the melting temperature of Fe₃S composition at 21 GPa (~1100 °C³). Thus, it is likely that the diamonds and their inclusions are formed at the same time or immediate aftermath of mantle solidification.

The thermal history of UPB is still under debate and robust conclusion on this matter requires extensive studies of ureilite samples and their silicate materials. Thus, it is beyond the scope of our paper which concentrates on the carbonic phases and their inclusions in Almahata Sitta MS-170 meteorite.

To address the issue, we have edited lines 188-189 which now reads:

L. 188-189: ...formation from a liquid phase at the same time or immediate aftermath (depending on the UPB's thermal history) of the solidification of the UPB...

After that, I have no further objections to the publication of this paper.

Thank you very much for your comments and for your concern to improve the manuscript.

References:

1. Goodrich, C. A. *et al.* Origin and history of ureilitic material in the solar system: The view from asteroid 2008 TC3 and the Almahata Sitta meteorite. *Meteorit. Planet. Sci.* **50**, 782–809 (2015).
2. Barrat, J.-A. *et al.* Partial melting of a C-rich asteroid: Lithophile trace elements in ureilites. *Geochim. Cosmochim. Acta* **194**, 163–178 (2016).
3. Fei, Y., Li, J., Bertka, C. M. & Prewitt, C. T. Structure type and bulk modulus of Fe₃S, a new iron-sulfur compound. *Am. Mineral.* **85**, 1830–1833 (2015).